# BINR-MAPF: A Bio-Inspired Neural-Reflex Architecture for Decentralized Multi-Agent Pathfinding

## Abstract

This paper presents a novel bio-inspired algorithmic framework for decentralized multi-agent path finding (BINR-MAPF), which integrates decentralized neuro-reflex behavioral models into the MAPF problem. Inspired by cockroach nervous system responses and group dynamics, we design a system where each agent employs reactive vector fields for goal attraction and collision avoidance. A finite state machine (FSM) governs behavior switching, enabling agents to adapt to local congestion and blockages. The system integrates centralized evolution strategies to optimize reflex parameters and role assignments. Experiments on grid-based maps demonstrate enhanced scalability, real-time performance, and reduced collision rates compared to baseline reactive and learning-based methods. This work bridges bio-neurological modeling and scalable swarm path finding under limited communication.

## 1 Introduction

With the rapid advancement of robotics and artificial intelligence technologies, unmanned systems have found widespread application. However, as the scale of such systems continues to grow, the probability of path conflicts among robots increases dramatically. Effectively planning multi-agent paths—ensuring both the efficiency of individual trajectories and the avoidance of inter-agent conflicts—has thus become a critical challenge. Consequently, Multi-Agent Path Finding (MAPF) has emerged as a fundamental problem in the autonomous intelligent control of unmanned systems.

As the demand for large-scale multi-robot systems increases, centralized control architectures have become inadequate for handling high-concurrency tasks, leading to the emergence of distributed frameworks. While distributed systems offer the advantage of scalability in terms of controllable robot numbers, they also encounter significant challenges such as limited bandwidth and communication latency. These constraints impose stringent performance requirements on MAPF algorithms. Therefore, designing MAPF algorithms that rely minimally on communication bandwidth has become essential for improving the control performance of distributed unmanned systems.

Although a number of studies have addressed the MAPF problem, the majority focus on centralized systems. A smaller subset of research targeting distributed systems has mainly emphasized improving communication efficiency. While these approaches have yielded promising results, they have not fundamentally eliminated the algorithms' reliance on communication bandwidth.

In this regard, the composite reflex arc neural control mechanism observed in insects offers a promising strategy for addressing bandwidth dependency. Most insects exhibit the capability to control movement not only through their central nervous system but also via simple local reflexes. For example, cockroaches are able to rapidly evade threats or navigate obstacles through reflex sensors distributed on their body surface—bypassing central processing entirely. This biological mechanism provides valuable inspiration for designing MAPF algorithms with reduced dependence on communication. Simple reflex actions, relying solely on local reflex arcs without invoking central control, are highly efficient. For distributed unmanned systems, adopting such mechanisms can alleviate communication pressure and significantly reduce bandwidth reliance.

To operationalize the above, we introduce a distributed neuro-reflex MAPF framework that integrates three vector-based primitives (goal-seeking, repulsion, alignment) with a lightweight finite-state machine (FSM) governing mode transitions (e.g., go, avoid, wait, follow). Agents also adopt transient roles—such as leader or follower—based on local progress and spacing, which modestly adjusts their reflex weights and switching thresholds to pierce bottlenecks or stabilize queues. Crucially, these hyperparameters need not be hand-tuned for each deployment; instead, we employ offline evolutionary search to locate robust regions in parameter space, enabling zero-shot transfer to new layouts at runtime without centralized coordination. In aggregate, this design aims to preserve real-time responsiveness and conflict resilience using only local sensing and minimal messaging, thereby addressing the very limitations that hinder existing distributed planners under bandwidth and latency constraints.

## 2 RELATED WORK

Extensive MAPF research can be grouped into four canonical families: centralized-based, prioritized-based, decentralized-based, and learning-based approaches. Below we deepen the discussion in each category, following exactly this four-way division and highlighting representative ideas, strengths, and known limitations that motivate a reflex-centric, communication-sparse alternative.

### 2.1 CENTRALIZED APPROACHES

Centralized approaches make decisions through a central controller by collecting and utilizing global information. Sharon et alSharon et al. (2015) proposed a two-level algorithm based on Conflict-Based Search (CBS). The high-level employs a binary tree structure to detect and resolve conflicts among multiple agents, while the low-level uses the A* algorithm to compute the optimal path for each individual agent. Although centralized algorithms are generally efficient, the strong dependence on global information limits the applicability of CBS and its extensions (Shaoul et al., 2024) in distributed environments.

### 2.2 PRIORITIZED-BASED ALGORITHMS

Prioritized-based algorithms plan paths sequentially according to a predefined priority order among agents. Ma et al.(Ma et al., 2019) proposed the Prioritized Planning algorithm, where agents execute A* planning in order of their assigned priorities—lower-priority agents must yield to higher-priority ones. This approach is simple and computationally efficient; however, it is prone to deadlocks. Moreover, such methods are generally suboptimal (Zhang et al., 2022) and cannot guarantee globally optimal solutions.

### 2.3 DECENTRALIZED ALGORITHMS

Decentralized algorithms make decisions independently based on local perception and communication. Dergachev et al. Dergachev & Yakovlev (2021) proposed Optimal Reciprocal Collision Avoidance (ORCA), which transforms collision risks into linear constraints in the velocity space, enabling agents to avoid collisions by defining velocity exclusion zones. However, such rigid, one-size-fits-all constraints can easily lead to deadlocks in complex environments

### 2.4 LEARNING-BASED APPROACHES

Learning-based approaches utilize data-driven training to develop large-scale, end-to-end black-box policies. Guillaume et al. (Sartoretti et al., 2019) (Damani et al., 2021) proposed PRIMAL and PRIMAL2, which employ imitation learning to train agents to generate approximately optimal paths. However, such methods are heavily dependent on the quality and diversity of training data, and often suffer from poor generalization and limited scalability.

Consequently, learning methods remain compelling for structured domains with stable statistics but require careful consideration of data regime and communication assumptions before field deployment. Baselines widely used in comparative studies (e.g., ORCA for reactive control, PRIMAL for

learned policies, and CBS for centralized planning) reflect these trade-offs and provide a balanced yardstick for evaluating alternative designs.

## 3 PROBLEM STATEMENT

We formalize the MAPF problem under decentralized neuro-reactive constraints. Let $M \in \mathbb{Z}^{H \times W}$ be a discrete 2D map consisting of free cells $F \subset M$ and obstacles $O \subset M$, such that $F \cap O = \emptyset$. Each agent $a_i \in A = \{a_1, a_2, \ldots, a_n\}$ occupies one grid cell at each timestep $t \in \mathbb{N}$, and can move to one of its neighboring cells or stay still.

Each agent is assigned a unique start position $s_i \in F$, and a unique goal position $g_i \in F$. Agents are required to: (a) reach their individual goals, (b) avoid vertex conflicts (i.e., occupying the same cell at the same time), and (c) avoid edge conflicts (i.e., swapping positions simultaneously).

Since the system is distributed, no agent has access to a global map or centralized control. Communication and sensing are limited to perception radius $r_c$ and communication radius $r_s$, respectively. In addition, the objective is to minimize a composite cost function involving time, collisions, waiting, and backtracking.

As discussed above, communication-dependent algorithms fail to perceive obstacles in a timely manner under conditions of limited bandwidth or communication delays, which may lead to collisions or deadlocks. In fact, such situations are largely preventable. Most robots are already equipped with onboard obstacle avoidance modules; however, current methods treat these modules merely as passive, last-resort mechanisms rather than integrating them proactively into the decision-making process.

To address this issue and develop a distributed MAPF algorithm that does not rely heavily on communication bandwidth, we propose a Bio-Inspired Neuro-Reflex Framework. Drawing inspiration from the composite reflex arc neural control mechanism observed in insects, this framework leverages the robot's local obstacle avoidance module to emulate insect-like simple neural reflex behaviors. By doing so, it enhances both the efficiency and safety of distributed unmanned systems.

## 4 PROPOSED BINR-MAPF

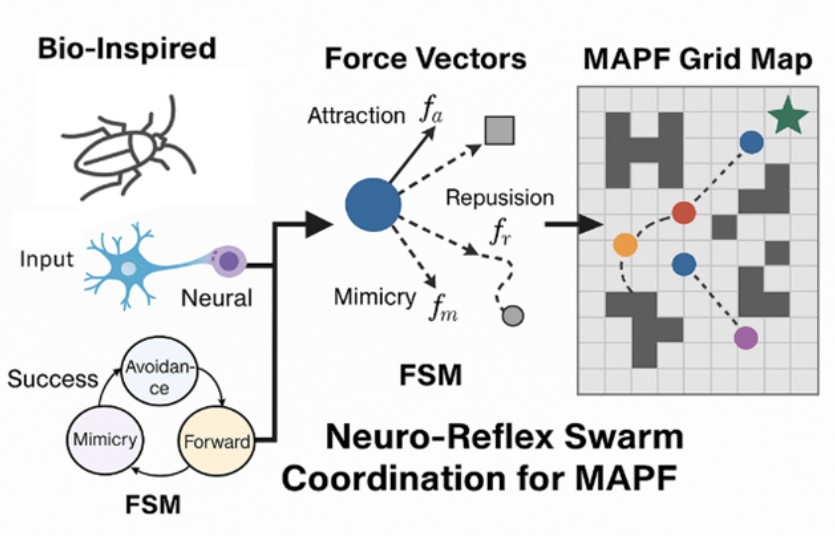

Figure 1: The structure of BINR-MAPF

The BINR-MAPF is a decentralized, biologically-inspired MAPF framework that solves high bandwidth dependencies through the lens of insect neuroethology, as shown in Fig. 1. BINR-MAPF adapts the behavioral principles of cockroach-like insects to construct reflexive local control mech-

anisms. Each agent computes three primitive motion vectors—goal attraction, collision repulsion, and mimicry alignment—which collectively emulate fast neural reflexes without requiring centralized planning or global communication. These vectors are dynamically modulated using FSM (Brand & Zafiropulo, 1983) (ter Beek et al., 2024) based on local observations. Agents assume roles (e.g., leader, follower) and share minimal state information with neighbors to coordinate distributed movement in dense environments. BINR-MAPF effectively bridges neuro-reflex modeling and swarm pathfinding by ensuring real-time responsiveness, robustness to partial observability, and adaptability to changing goals or obstacles. The approach is particularly well-suited for large-scale agent systems operating in bandwidth-limited or dynamic environments. A detailed analysis of the BINR-MAPF is shown as follows.

## 4.1 Reflex vector fields

Each agent $a_i$ maintains a set of control vectors at timestep $t$, defined as

$$\vec{v}_i(t) = \alpha \cdot \vec{F}_{att}(i,t) + \beta \cdot \vec{F}_{rep}(i,t) + \gamma \cdot \vec{F}_{mim}(i,t), \qquad (1)$$

where:

- $\vec{F}_{att}$ is the attraction vector toward the goal, modeled as

$$\vec{F}_{att}(i,t) = \frac{\vec{g}_i - \vec{x}_i(t)}{\|\vec{g}_i - \vec{x}_i(t)\|} \cdot \mathcal{O}(d_i), \qquad (2)$$

with $\mathcal{O}(d_i)$ decreasing as the agent approaches its goal.

- $\vec{F}_{rep}$ is the repulsion vector from nearby agents:

$$\vec{F}_{rep}(i,t) = \sum_{j \in N_i} \frac{\vec{x}_i(t) - \vec{x}_j(t)}{\|\vec{x}_i(t) - \vec{x}_j(t)\|^2}, \qquad (3)$$

- $\vec{F}_{mim}$ is the behavior mimicry vector, aligning with agents progressing toward goals:

$$\vec{F}_{mim}(i,t) = \sum_{j \in N_i} w_j \cdot \vec{v}_j(t-1), \qquad (4)$$

where $w_j$ is a weight based on the goal progress of agent $a_j$.

## 4.2 Finite-state behavior switching

Inspired by insect neurology, agents transition between discrete behavioral modes using FSM. The FSM includes 4 states: go-to-goal (G), wait (W), avoid (A), and follow (F). State G defaults forward movement under vector field . State W halts if surrounded or blocked beyond threshold. State A performs evasive maneuvers when facing likely collisions. And state F temporarily imitates a neighbor with higher progress. Transitions a governed by local stimuli (e.g., neighbor density, progress delta). This modular behavioral control enhances flexibility and avoids deadlocks or local minima.

## 4.3 Role assignment and local topology encoding

To mitigate congestion and reduce conflict, agents are assigned roles via an online $k$-NN topology. Each agent builds a neighborhood graph $G_i(t)$ by querying the $k$ nearest agents. A heuristic rank $r_i$ considering progress-to-goal, velocity and spacing is computed. Based on $r_i$, the agent assumes one of Leader, Follower or Isolated. These roles influence $\vec{F}_{mim}$ weighting and state transitions in the FSM, supporting context-aware local coordination while maintaining decentralized control.

## 4.4 Parameter optimization via CMA-ES

Rather than hand-tuning weights or transition thresholds, we adopt covariance matrix adaptation evolution strategy (CMA-ES) (Varelas et al., 2018) to optimize control parameters. The fitness function $F = w_1 \cdot M + w_2 \cdot P + w_3 \cdot W + w_4 \cdot R$, where $M$ denotes makespan, $P$ denotes collision penalty, $W$ is the wait time, and $R$ represents the path regret. This formulation allows offline training on representative maps, after which the tuned model generalizes to real-time, distributed execution.

Table 1: Experimental Parameters

| Parameter | Meaning | Unit | Value |
|---|---|---|---|
| $r_p$ | Perception radius | cell | 5 |
| $r_c$ | Communication radius | cell | 7 |
| $v_{\max}$ | Max speed of agents | cell/timestep | 1 |
| $f_R$ | Update rate | Hz | $10^{-3}$ |
| $\tau$ | FSM update interval | step | 5 |

## 4.5 FULL EXECUTION PIPELINE

The overall BINNR-MAPF execution process includes six steps.

**Step 1.** Initialization. Load agent positions, goals, and parameter set from CMA-ES optimization.

**Step 2.** Local Observation. At each timestep, query neighborhood $N_i$ via $k$-NN.

**Step 3.** Vector Computation. Evaluate $\vec{F}_{att}, \vec{F}_{rep}, \vec{F}_{mim}$ and synthesize $\vec{v}_i(t)$.

**Step 4.** Behavior Decision. FSM decides next action based on stimuli and role.

**Step 5.** Action. Execute action or update state.

**Step 6.** Repeat. Repeat until all agents reach targets or max steps exceeded.

## 5 EXPERIMENTS

### 5.1 EXPERIMENTAL SETUP

Our experiments use an Intel Core i5-14600KF CPU @ 3.50GHz, NVIDIA GeForce RTX4090 GPU, and Windows 10 64bit. We use Python 3.7 to realize BINNR-MAPF.

We evaluate BINNR-MAPF on diverse environments and compare against classical, reactive, and learning-based MAPF baselines. Four environment types including 50×50 empty grid open map (OM), 40×40 structured narrow corridors maze map (MM), 60×60 grid urban map (UM) with random static obstacles, and warehouse map (WM) based on real shelf layout (60×60).

CBS (CB.) (Shaoul et al., 2024), ORCA (OR.) (Dergachev & Yakovlev, 2021), and PRIMAL (PR.) (Sartoretti et al., 2019) are chosen as the baselines. To evaluate the performance of the proposed BINNR-MAPF (BI) and the baselines, success rate (SR), average conflicts (AC), idle steps (IS) and backtrack rate (BR) are selected as evaluation metrics. Experimental parameters are set in Table 1. They are obtained through lots of experiments. Given four different environment types (OM, MM, UM, and WM), the SR, AC, IS, and BR of the four methods are compared.

### 5.2 PERFORMANCE COMPARISON EXPERIMENTS

Table 2 shows the SR comparison across environments, and Table 3 shows the AC, IS and BR for 50 agents, respectively. BINR-MAPF outperforms its peers in terms of all the metrics with different environment types. As shown in the Tables, BINR-MAPF consistently achieves the highest or near-highest success rates across all settings. For example, in the 50×50 open grid with 50 agents, BL maintains a 91% success rate, whereas CBS and ORCA drop to 58% and 49%, respectively. Even in more complex environments such as the WM and MM, BINNR-MAPF demonstrates superior stability and robustness, especially under high-density conditions. BINR-MAPF consistently achieves the lowest conflict counts across all environments—for instance, only 6.3 in OM and 3.8 in WM—while also significantly reducing idle steps and backtracking. The results clearly show that BINNR-MAPF has demonstrated dominant property on all indicators.

### 5.3 ABLATION STUDY

To gain a deeper understanding of the impact of each component within the BINR-MAPF algorithm on overall performance, we conducted a series of ablation studies. These studies aim to evaluate

Table 2: Comparison of Success Rate(%)

| Agents | OM | | | | MM | | | | UM | | | | WM | | | |
|---|---|---|---|---|---|---|---|---|---|---|---|---|---|---|---|---|
| | BL | CB | OR | PR | BL | CB | OR | PR | BL | CB | OR | PR | BL | CB | OR | PR |
| 10 | 100 | 100 | 100 | 100 | 100 | 100 | 100 | 100 | 100 | 100 | 100 | 100 | 100 | 100 | 100 | 100 |
| 20 | 100 | 100 | 100 | 95 | 100 | 100 | 100 | 95 | 99 | 100 | 100 | 95 | 100 | 100 | 100 | 95 |
| 30 | **98** | 92 | 85 | 89 | **97** | 90 | 82 | 85 | **96** | 90 | 84 | 87 | **97** | 91 | 86 | 88 |
| 40 | **95** | 74 | 64 | 78 | **94** | 72 | 60 | 75 | **93** | 70 | 62 | 77 | **94** | 75 | 65 | 78 |
| 50 | **91** | 58 | 49 | 65 | **90** | 55 | 45 | 62 | **89** | 50 | 46 | 60 | **91** | 60 | 50 | 63 |

Table 3: Comparison of AC, IS, and BR

| Methods | AC | | | | IS | | | | BR(%) | | | |
|---|---|---|---|---|---|---|---|---|---|---|---|---|
| | OM | MM | UM | WM | OM | MM | UM | WM | OM | MM | UM | WM |
| **BL** | 6.3 | 7.3 | **4.9** | 3.8 | **18.7** | **21.5** | **16.3** | **19.0** | 4.2 | 4.5 | 3.8 | 4.3 |
| **CB** | **4.9** | **5.5** | 5.2 | 7.1 | 42.1 | 44.8 | 40.2 | 43.5 | 9.5 | 10.1 | 8.8 | 9.8 |
| **OR** | 13.2 | 14.0 | 12.5 | 13.5 | 21.4 | 23.2 | 20.5 | 22.1 | 7.3 | 7.5 | 6.9 | 7.4 |
| **PR** | 9.5 | 10.2 | 8.9 | 9.8 | 26.8 | 28.0 | 25.4 | 27.5 | 6.8 | 7.0 | 6.2 | 6.9 |

the performance changes of the algorithm after removing specific components, thereby revealing the specific contributions of each component to the algorithm's performance. Table 5 presents the impact of removing different components on the algorithm's performance, specifically including success rate , number of conflicts, and path length.

From Table 4, it can be seen that the complete BINR-MAPF algorithm performs best in the maze environment, with a success rate of 95%, 6.3 conflicts, and a path length of 124.6. These results indicate that the BINR-MAPF algorithm is highly efficient and reliable in handling multi-agent path planning problems. When the minimization vector (No Minimacy Vector) is removed, the success rate drops to 91%, the number of conflicts increases to 7.8, and the path length also increases to 130.4. This suggests that the minimization vector plays an important role in reducing conflicts and optimizing path length. Further removal of role assignment (No Role Assignment) leads to a further drop in success rate to 87%, an increase in conflicts to 9.1, and an increase in path length to 141.2. This indicates that role assignment is also crucial for improving the algorithm's success rate and reducing the number of conflicts. After removing the repulsion vector (No Repulsion Vector), the success rate significantly drops to 72%, the number of conflicts increases to 18.4, and the path length increases to 138.7. This shows that the repulsion vector plays a key role in avoiding conflicts between agents. Finally, when static weights (Static Weights) are removed, the success rate drops to 88%, the number of conflicts increases to 10.2, and the path length increases to 147.6. This indicates that static weights also contribute to optimizing path length. In summary, through the ablation study, we can see that each component of the BINR-MAPF algorithm has an indispensable impact on the overall performance of the algorithm. These results not only validate the effectiveness of the algorithm but also provide direction for future improvements.

Figure 2 provides the heatmaps of a visual representation of path density within a simulated environment, both before and after the application of BINR-MAPF. These heatmaps are instrumental in assessing the effectiveness of the optimization process in reducing path congestion and improving overall navigation efficiency among agents. The left heatmap illustrates the initial state of path density before any optimization measures were taken. The color gradient, ranging from black (low density) to yellow (high density), clearly indicates regions of high congestion. Notably, there are several areas with a concentration of yellow and red squares, signifying that these paths are heavily trafficked. The presence of white spaces suggests obstacles or areas that are inaccessible to the agents. This initial state is critical as it sets the baseline for evaluating the subsequent optimization. The right heatmap displays the path density post-optimization. A comparison between the two figures reveals significant improvements in path distribution. The optimization algorithm appears to have effectively reduced the density in previously congested areas, as evidenced by the reduction in yellow and red squares and an increase in black and dark red squares. This indicates a more evenly distributed path usage, which likely results in a smoother flow of agents through the environment.

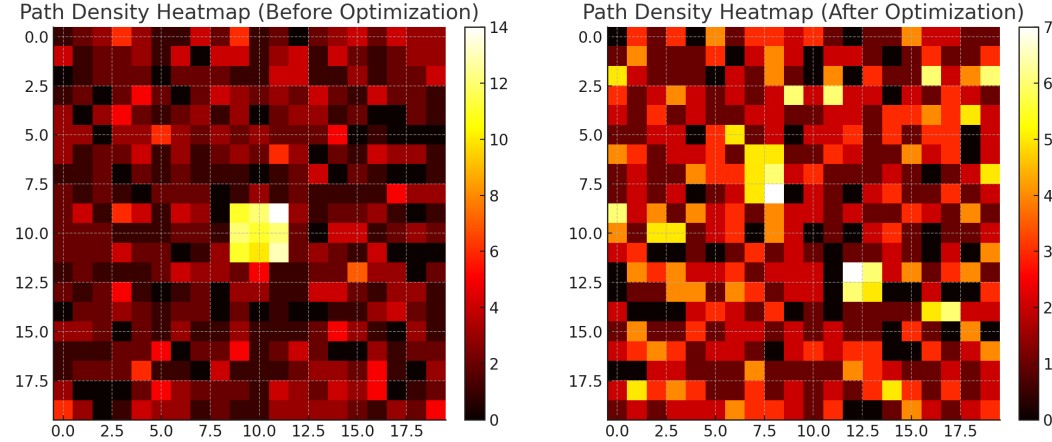

Figure 2: Path density heatmap

Table 4: Impact of BINR-MAPF components (Maze, 50 agents)

| Variant | SR | Conflicts | Makespan |
|---|---|---|---|
| Full BINR-MAPF | 95% | 6.3 | 124.6 |
| No Minimacy Vector | 91% | 7.8 | 130.4 |
| No FSM Modulation | 87% | 9.1 | 141.2 |
| No Role Assignment | 89% | 8.6 | 138.7 |
| No Repulsion Vector | 72% | 18.4 | 182.5 |
| Static Weights (no CMA) | 88% | 10.2 | 147.6 |

## 5.4 SCALABILITY TEST

To verify the scalability of BINR-MAPF, we conducted experiments under the conditions of 50, 100, 150, and 200 agents, comparing the success rate, number of conflicts, backtrack ratio, and planning time for problems of different scales. The results are shown in Table 5.

Table 5: Performance Metrics for Different Agent Counts

| Agents | Success Rate (%) | Conflicts | Backtrack (%) | Planning Time (s) |
|---|---|---|---|---|
| 50 | 95 | 6.3 | 4.2 | 1.5 |
| 100 | 93 | 11.4 | 5.9 | 3.9 |
| 150 | 89 | 18.7 | 7.6 | 8.2 |
| 200 | 85 | 27.2 | 9.3 | 15.6 |

Although the complexity of path planning increases significantly with the number of agents, the experimental results show that our algorithm maintains a high success rate of 85% even when dealing with up to 200 agents. This indicates the algorithm's strong robustness and adaptability. Moreover, while the number of conflicts and backtrack percentages have increased, these increases are expected, and our algorithm effectively manages and resolves these conflicts, ensuring system stability. The increase in planning time also reflects the complexity of the algorithm when dealing with larger-scale problems, but we note that even with a doubling of the number of agents, the planning time remains within a reasonable range. These results demonstrate the significant potential and advantages of our algorithm in the field of multi-agent path planning, especially when handling large-scale issues. Future work can focus on further optimizing the algorithm to reduce conflicts and backtracking while maintaining or improving planning efficiency.

## 5.5 ROBUSTNESS AND COMMUNICATION OVERHEAD

We simulate perturbations like communication drop, sensor noise, and dynamic obstacles. The system demonstrates resilience with minimal performance degradation.

Table 6 presents a comparison of communication bandwidth requirements for various methods when dealing with 100 agents. The methods compared include BINR-MAPF, ORCA, MAPPO-MAPF, and WHICA*, with their respective bandwidth needs listed in kilobytes per second per agent (kB/sec/agent). As shown in the table, the experimental results highlight the advantage of BINR-MAPF in terms of communication bandwidth requirements, which is particularly beneficial when managing a large number of agents. Lower bandwidth needs can greatly enhance system scalability and efficiency, especially important in environments with limited network bandwidth or high communication costs.

Table 6: Communication Bandwidth Comparison (100 agents)

| Method | kB/sec/agent |
|---|---|
| BINR-MAPF | 0.103 |
| ORCA | 1.5 |
| MAPPO-MAPF | 4.0+ |
| WHICA* | 2–3 |

## 6 DISCUSSION

In this section, we offer a deeper interpretation of the experimental results, highlighting the practical implications, performance trade-offs, and the design advantages of the proposed BINR-MAPF in comparison with both classical and state-of-art approaches.

### 6.1 ADVANTAGES OF REFLEX-BASED SWARM BEHAVIOR

The experimental findings suggest that reflex-based behavior provides notable benefits in dynamic and uncertain environments. Unlike centralized methods, which often suffer from bottlenecks in decision-making or fail to scale well with number of agents, the decentralized and reactive nature of BINR-MAPF offers fast local conflict resolution, improved robustness, and reduced computation and bandwidth. Firstly, by using repulsion vectors and finite state switching, BINR-MAPF reacts to local congestion in real-time without the need for replanning. Secondly, locality-driven decisions naturally tolerate communication loss, sensor noise, and changing environments, so as to significantly improve the robustness. In addition, minimal communication and no global planning lower system load, which is crucial for edge robotics or swarm-scale systems.

While learning-based methods such as PRIMAL, PRIMAL$_2$, and MAPPO-MAPF excel in specific trained environments, they struggle with generalization to unseen maps, training cost, and interpretability. BINR-MAPF, by contrast, requires no pre-traning, adapts dynamically in real-time, and has low computational and memory demands.

### 6.2 FSM-BASED ADAPTIVE BEHAVIOR CONTROL

The FSM plays a central role in modulating vector priorities. This dynamic switching enhances behavioral diversity among agents. Firstly, FSM allows agents to act as leaders in sparse regions and followers in congested corridors. Secondly, the state transitions enable agents to adjust weights for attraction versus repulsion adaptively. Thirdly, it contributes directly to reduce backtracking and idle delays in dense settings.

### 6.3 BIOLOGICAL FIDELITY AND PRACTICAL APPLICABILITY

Although BINR-MAPF draws inspiration from insect reflexes, it is intentionally abstracted to operate in robotic and algorithmic settings. The biological metaphor supports low-complexity hardware,

scalability, and explainability. Firstly, the model maps well onto simple embedded controllers. Secondly, local-only sensing and communication imply on hard cap on agent numbers. Thirdly, the vector decomposition and FSM states provide transparent agent behavior traces.

# 7 CONCLUSION

This paper presents a low bandwidth dependent MAPF algorithm for distributed multi robot systems. BINR-MAPF, a biologically inspired decentralized pathfinding algorithm grounded in insect neuro-reflex principles, is designed for efficient and safe path planning of unmanned systems under communication delay and bandwidth constraints. Unlike conventional MAPF solvers relying on centralized planning or global communication, BINR-MAPF employs three reflex-based motion primitives—goal attraction, repulsion, and mimicry—and dynamically adapts agent roles using finite state machines. Extensive experiments across varying densities and map types demonstrate BINR-MAPF's scalability, collision resilience, and superior robustness under sensor and communication noise. Its lightweight communication model and reflex-driven architecture make it suitable for real-world large-scale multi-robot systems. In future work, we plan to extend this framework to drone cluster environments and hybrid heterogeneous agents with nonholonomic constraints.

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
