# OpenReview forum: "BINR-MAPF: A Bio-Inspired Neural-Reflex Architecture for Decentralized Multi-Agent Pathfinding"
_ICLR.cc/2026/Conference — ICLR 2026 Conference Withdrawn Submission_

### Official Review · Reviewer_j1vZ · 2025-10-31

**Soundness:** 1
**Presentation:** 2
**Contribution:** 1
**Rating:** 2
**Confidence:** 4

**Summary:**

The paper presents BINR-MAPF, a novel approach for decentralized MAPF, where each agent has a set of control vectors: goal attraction, collision repulsion, and mimicry alignment (based on analogies to behavioral principles of insects). This behavior is modeled by an fenite states machines, and to select the roles of agents, an online KNN method is used. The parameters of such a decentralized system are adjusted using CMA-ES for each new scenario.

**Strengths:**

- The biological idea seems novel. The ranking of agents reminds me of how PIBT works, and there could be interesting parallels or ways to improve PIBT.

**Weaknesses:**

W1: The approach itself seems rather weak compared to existing large-scale imitation learning and hybrid methods. Using an FSM alone is unlikely to achieve competitive results. Still, outperforming PRIMAL on some maps is commendable.

W2: The related work section lacks coverage of existing baselines. I recommend consulting the recent review on ML for MAPF [1] and the POGEMA benchmark [2]. Similar to centralized approaches, there are many strong baselines (e.g., LaCAM* [3], LNS2+RL [4]), along with high-speed decentralized non-learnable solvers (e.g., PIBT [5]).

W3: Likewise, the paper omits comparisons with many modern decentralized learnable approaches. I recommend considering baselines discussed in the recent MAPF-GPT paper [6]. A comparison with PIBT would also be highly valuable. Comparing only with discrete ORCA, CBS, and PRIMAL is not sufficient.

W4: The paper does not provide any code and omits the reproducibility checklist. Anyone attempting to reproduce the results would face significant difficulties.

W5: The experimental results are hard to follow. Both the scenarios and algorithms are abbreviated (e.g., PR for PRIMAL, WM for Warehouse Map). I suggest that the authors include full names or at least move the abbreviations’ explanations to the table descriptions.

W6: I found no solid support for the claim that the approach can operate in real time, as stated several times in the paper. I’m not sure whether subsection 5.4 (Scalability Test) fully addresses this question. Moreover, for modern learnable approaches, handling hundreds of agents in a single map is common, so evaluating only up to 200 agents seems insufficient.

[1] Alkazzi JM, Okumura K. A comprehensive review on leveraging machine learning for multi-agent path finding. IEEE Access. 2024 Apr 22;12:57390-409.

[2] Skrynnik A, Andreychuk A, Borzilov A, Chernyavskiy A, Yakovlev K, Panov A. POGEMA: A Benchmark Platform for Cooperative Multi-Agent Pathfinding. In The Thirteenth International Conference on Learning Representations.

[3] Okumura K. Engineering LaCAM*: Towards Real-time, Large-scale, and Near-optimal Multi-agent Pathfinding. InProceedings of the 23rd International Conference on Autonomous Agents and Multiagent Systems 2024 May 6 (pp. 1501-1509).

[4] Wang Y, Duhan T, Li J, Sartoretti G. LNS2+ RL: Combining multi-agent reinforcement learning with large neighborhood search in multi-agent path finding. InProceedings of the AAAI Conference on Artificial Intelligence 2025 Apr 11 (Vol. 39, No. 22, pp. 23343-23350).

[5] Okumura K, Machida M, Défago X, Tamura Y. Priority inheritance with backtracking for iterative multi-agent path finding. Artificial Intelligence. 2022 Sep 1;310:103752.

[6] Andreychuk A, Yakovlev K, Panov A, Skrynnik A. Mapf-gpt: Imitation learning for multi-agent pathfinding at scale. InProceedings of the AAAI Conference on Artificial Intelligence 2025 Apr 11 (Vol. 39, No. 22, pp. 23126-23134).

**Questions:**

Q1: Could you clarify how the FSM states and transitions were designed?

Q2: How sensitive is the approach to the CMA-ES parameter tuning process? Would the performance degrade significantly if CMA-ES is not run for a specific scenario?

---

### Official Review · Reviewer_tfsi · 2025-10-31

**Soundness:** 3
**Presentation:** 3
**Contribution:** 3
**Rating:** 4
**Confidence:** 2

**Summary:**

The paper proposes BINR-MAPF, a biologically inspired decentralized multi-agent pathfinding algorithm that draws on insect neuro-reflex principles. Each agent computes local control vectors for goal attraction, repulsion, and alignment, modulated by a finite state machine (FSM) governing behaviors like go, avoid, wait, and follow. Role assignments (leader/follower) further adapt vector weights, while parameters are optimized offline via CMA-ES. The approach aims to achieve scalable pathfinding under limited communication. Experiments on grid maps and warehouse layouts show higher success rates and lower collisions than CBS, ORCA, and PRIMAL. Ablation studies and scalability tests are also included.

Overall, it’s an interesting biologically inspired take on decentralized MAPF, though a bit empirical and engineering-heavy rather than theoretically novel.

**Strengths:**

Strong motivation grounded in distributed robotics challenges (latency, bandwidth). The reflex-based local control is conceptually clean and scalable. FSM-based adaptive behavior switching and leader-follower roles provide flexibility. Empirical results are broad and show consistent gains. Communication efficiency analysis (Table 6) is a plus, rarely reported in MAPF papers.

**Weaknesses:**

The 'bio-inspired' claim is mostly metaphorical; there’s no real biological modeling. No theoretical analysis of convergence, stability, or collision guarantees. Experiments are simulation-only with hand-picked environments. Statistical rigor is missing, and parameter tuning via CMA-ES raises questions about generalization. Some claims (e.g., zero-shot transfer) are overstated given purely offline training. Writing style occasionally awkward and figures not polished.

**Questions:**

How does BINR-MAPF behave under delayed sensing or partial observability (beyond the small perturbation test)?
Could CMA-ES parameters be replaced by online adaptive tuning?
How is deadlock formally prevented, FSM switching doesn’t guarantee completeness.
Is there a limit on density or environment complexity before performance drops sharply?

---

### Official Review · Reviewer_FvtV · 2025-11-03

**Soundness:** 2
**Presentation:** 1
**Contribution:** 1
**Rating:** 2
**Confidence:** 3

**Summary:**

The paper introduces BINR-MAPF, a decentralized multi-agent pathfinding framework inspired by insect neuro-reflex systems. Drawing from cockroach behaviors, each agent operates using local reactive vector fields for goal attraction and obstacle avoidance, while a finite state machine governs behavior switching (go to goal (G), wait (W), avoid (A), and follow (F)) in response to congestion or blockages.The authors claim the method bridges biological reflex principles with swarm pathfinding and provides better coordination among agents.

However, this paper is not well-written. There is no proper literature review or experiments. For such a well-studied problem, the paper has only 10 references. I suggest the authors consider improving the paper by performing proper literature review and experimental comparison with other MAPF papers.

**Strengths:**

- There is some novelty in considering biological behaviors.

**Weaknesses:**

- It is not clear how this approach compares with many other bio-inspired algorithms (ant colony, particle swarm, etc.)

**Questions:**

Why is this method better than state of the art in the MAPF literature?

---

### Official Review · Reviewer_cZUm · 2025-11-04

**Soundness:** 3
**Presentation:** 3
**Contribution:** 3
**Rating:** 6
**Confidence:** 4

**Summary:**

This paper introduces BINR-MAPF, a bio-inspired decentralized multi-agent pathfinding framework that mimics insect neuro-reflex behaviors using reactive vector fields and finite-state control to achieve scalable, low-bandwidth, and collision-free coordination. The key contributions are:

1. A neural-reflex inspired decentralized MAPF model combining vector fields, FSM-based adaptation, and evolutionary optimization.

2. Demonstration of strong scalability (up to 200 agents) and robustness under communication noise and sensor perturbations.

3. A bridge between biological motor control concepts and algorithmic swarm pathfinding, emphasizing low-bandwidth, real-time control.

**Strengths:**

Originality is strong. The bio-inspired framing of multi-agent pathfinding through neural reflexes and FSM-based behavior modulation is creative. While reflex-based local control has precedents in robotics, integrating it with CMA-ES parameter search and distributed role-switching for MAPF is novel. The idea of combining biological realism with evolutionary tuning for decentralized scalability is particularly fresh and well-motivated.

Quality is good. The paper is technically coherent, clearly describing the components—vector fields, FSM, and CMA-ES optimization. Experiments are extensive: multiple maps, baselines (CBS, ORCA, PRIMAL), ablation analysis, scalability up to 200 agents, and robustness tests under noise. Results consistently show superior performance across metrics such as success rate, conflicts, idle steps, and backtracking, which supports the claimed advantages. However, there is no statistical significance analysis (no variance/error bars), runtime comparisons focus on planning time but not total computation or scalability trends beyond 200 agents, and there is no real-world validation (simulation only). Still, the methodology is sound and well-executed for a conference paper.

Clarity is above average. The paper is logically structured, with clear sectioning and consistent mathematical notation, and its figures—including the FSM diagram and heatmaps—effectively illustrate key ideas. Nonetheless, there are areas for improvement: figure quality could be enhanced (particularly in resolution and labeling); furthermore, summarizing the FSM transition rules and role-switching heuristics in pseudocode would boost clarity; and while the acronym "BINR-MAPF" is somewhat cumbersome, maintaining consistent use of it would improve overall readability.

Significance is high. The problem—scalable decentralized MAPF under communication constraints—is timely and relevant to swarm robotics and distributed AI. The presented method offers a lightweight alternative to learning-heavy or communication-heavy approaches.
Its robustness, simplicity, and generalization without retraining make it attractive for real-world deployments (e.g., warehouse robots, UAV swarms). If validated further, it could inspire a new class of bio-inspired reflexive control systems for MARL and MAPF.

**Weaknesses:**

Lack of theoretical grounding or convergence analysis

Suggestions:
1. Provide an analytical discussion (even qualitative) of conditions ensuring conflict-free convergence under the reflex vector composition.
2. Reference existing work on potential fields or velocity obstacles to position the reflex model theoretically.

Limited insight into FSM design and switching dynamics

Suggestions:
1. Include a transition table or pseudocode describing triggers (e.g., density > threshold → AVOID).
2. Visualize how switching frequency varies across environments (e.g., histogram of state transitions).

Evaluation missing broader baselines and metrics

Suggestions:
1. Add a modern learning-based MAPF baseline (MAPPO or Graph-Nets) for a stronger empirical contrast.
2. Report average step time or frame rate to validate real-time claims.

Scalability and generalization discussion could be deeper

Suggestions:
1. Analyze communication complexity O(n) or O(k) and demonstrate empirical scaling with n.
2. Discuss how reflex interactions behave when agent density saturates (e.g., >300 agents).

Minor presentation and reproducibility issues

Suggestions:
1. Include hyperparameter values and CMA-ES settings (population size, iterations).
2. Clarify how “leader/follower” roles are determined (formula for ranking ri).
3. Provide code or pseudocode to ensure reproducibility.
4. Improve figure captions and ensure consistent units (e.g., “cell/timestep”).

**Questions:**

1. What specific sensory thresholds trigger transitions between go/avoid/wait/follow? Are they learned or fixed after CMA-ES tuning?

2. Are there cases where reflex interactions create oscillations or deadlocks? If so, how are they mitigated?

3. Does a single optimized parameter set generalize across all maps, or is retraining needed?

4. How does runtime grow beyond 200 agents? Could this framework scale to 1,000+ agents on sparse maps?

5. How were CBS/ORCA/PRIMAL tuned to ensure fair runtime and bandwidth comparisons?

---

### Official Review · Reviewer_Kj9r · 2025-11-04

**Soundness:** 2
**Presentation:** 1
**Contribution:** 3
**Rating:** 4
**Confidence:** 3

**Summary:**

In this paper, the authors propose a simple but effective biologically-inspired decentralized MAPF algorithm. The authors show that a set of primitive agent behaviors, each controlled by a finite-state machine based on local information, is enough to solve many tasks that are becoming challenging for classical algorithms with the growth of agent counts. Though this work is an intriguing example of the application of swarm intelligence in practice and suggests itself as a SOTA, the algorithm’s description and overall presentation are not sufficiently good for publication yet.

Overall, the results are promising, but the presentation is poor. I believe that most of my concerns can be addressed during the rebuttal phase, and I am keen to increase my score if the presentation and discussion parts are improved.

**Strengths:**

*   Simple, interpretable swarm mechanics in use
*   Highly decentralized agent interaction with minimum communication
*   The algorithm is potentially lightweight in terms of computational resources
*   Better scalability with the number of agents in comparison to some other MAPF methods

**Weaknesses:**

1.  The Related Work section seems to be quite outdated and very limited. MAPPO-MAPF is mentioned but not cited. What is the connection to other swarm-based algorithms (if any), for example, to this one: https://www.nature.com/articles/s41598-025-88145-7?
2.  The choice of baselines is not justified. If deep RL baselines are not suitable for this setup, it should be explained in more detail, and the differences between the setups should be underlined. In any case, it would also be good to show the performance of SOTA baselines for reference, even for fixed maps.
3.  No limitations are discussed. What about maps that contain bugtrap patterns and similar? The proposed method seems to be quite shortsighted. Where do we need more intelligent behavior? Or is swarm intelligence all we need?
4.  The experimental setup is not sufficiently described. Do you use the same weights for every map type? How many maps are tested for each type? How do you choose initial positions and goals? Why are there no confidence intervals reported for the metrics?
5.  The formal description of the method is not sufficiently detailed (see questions).

**Questions:**

1.  The FSM module of the proposed algorithm should be described in more detail. What observations (stimuli) are used? How do they influence transitions? How does the FSM influence the contribution of different forces (please, explain more formally)?
2.  The training for the FSM is also opaque. How do you choose maps and tasks for tuning? Are they the same as for testing? How many trials are required for tuning the FSM?
3.  What is the “distributed environment”? (line 81)
4.  Line 205: How is the heuristic rank calculated?
5.  Mixing abbreviations for the proposed method: line 250 - BI and BL in tables.
6.  Ideally, metrics should be explained.
7.  Table 5 is misreferenced in line 292.
8.  What is the “minimization vector” (line 298)? Did you mean "mimicking"?
9.  Figure 2: What map is used? What is "before" and “after optimisation”? How are these densities formed?
10. Line 381: “system demonstrates resilience” - missing experimental results.
11. Table 6: How is it computed?
12. Line 418: “BINR-MAPF, by contrast, requires no pre-training,” - but what about FSM tuning?
13. Please fix numerous grammar mistakes and typos. The sentence “local-only sensing and communication imply on hard cap on agent numbers” (line 433) is nonsensical.

---

### Note · Authors · 2025-12-10

I have read and agree with the venue's withdrawal policy on behalf of myself and my co-authors.